**Technical Note : A universal method for measuring the thickness of microscopic calcite crystals, based on Bidirectional Circular Polarization**

Luc Beaufort*, Yves Gally*, Baptiste Suchéras-Marx*, Patrick Ferrand**, Julien Duboisset**

*Aix Marseille Univ, CNRS, IRD, INRAE, Coll. France, CEREGE, Aix-en-Provence, France
** Aix Marseille Univ, CNRS, Centrale Marseille, Institut Fresnel, Marseille, France

## 1.  Abstract

Coccoliths are major contributors to the particulate inorganic carbon in the ocean that is a key part of the carbon cycle. The coccoliths are few microns in length and weigh a few picograms. Their birefringence characteristics in polarized optical microscopy has been used to estimate their mass. This method is rapid and precise because camera sensors produce excellent measurement of light. However, the current method is limited because it requires a precise and replicable set up and calibration of the light in the optical equipment. More precisely, the light intensity, the diaphragm opening, the position of the condenser, and the exposure time of the camera have to be strictly identical during the calibration and the analysis of calcite crystal. Here we present a new method that is universal in the sense that the thickness estimations are independent from a calibration but results from a simple equation. It can be used with different cameras and microscope brands. Moreover, the light intensity used in the microscope does not have to be strictly and precisely controlled. This method permit to measure crystal thickness up to 1.7 µm. It is based on the use of one left circular polarizer and one right circular polarizer with a monochromatic light source using the following equation:

$$d = \frac{\lambda}{\pi \Delta n} arctan\left(\sqrt{\frac{I_{LR}}{I_{LL}}}\right)$$

where $d$ is the thickness, $\lambda$ the wavelength of the light used, $\Delta n$ the birefringence, $I_{LR}$ and $I_{LL}$ are the light intensity measured with a right and a left circular polarizer. Because of the alternative and rotational motion of the quarter-wave plate of the circular polarizer, we coined the name of this method 'Bidirectional Circular Polarization' (BCP).

## 2.   Introduction

Coccolithophores are abundant oceanic single cell algae that produce calcite plate called coccoliths, that are displayed around the cell to form an exoskeleton. Coccolithophores are extremely abundant in all ocean (Okada and Honjo, 1973) and some species form blooms that are detected by satellite imagery (Holligan et al., 1993). The coccoliths are major contributors to the particulate inorganic carbon (i.e., PIC) in the pelagic ocean (Milliman and Droxler, 1996;Suchéras-Marx and Henderiks, 2014).  that is a key part of the carbon cycle. They are important contributors to the carbonate counter pump (Ridgwell and Zeebe, 2005) and they are considered as climate stabilizer on long time scales (Zeebe and Westbroek, 2003;Höning, 2020). The calcite mass of the coccolith is therefore a parameter that is important to estimate for example to monitor the effect of ocean acidification on calcification (e.g. Beaufort et al., 2007;Beaufort et al., 2011) or to calculate their flux to the seafloor (Beaufort and Heussner, 1999). The coccoliths are so minute (few microns in length) and light (few picograms) that they can be weighed individually only with extreme labor and expensive equipment (Hassenkam et al., 2011;Beuvier et al., 2019). Alternatively, the birefringence characteristics of coccoliths in polarized optical microscopy have been used to estimate their mass (Beaufort, 2005;Beaufort et al., 2014;Bollmann, 2014;Fuertes et al., 2014). The justification for measuring birefringence is that it directly relates the color (and brightness) of a crystal observed under cross-polarized light microscopy to its thickness. The conversion comes without having to manipulate the particule. Moreover, this method is rapid and precise. The camera sensor produces excellent measurement of the light that travels through the polarizers and a calcite crystals which is converted into a thickness value, and mass when it is associated with the surface measurement. The thickness estimation made by this method has been recently positively evaluated by the independent measurements made by X-ray tomography at the European Synchrotron Radiation Facility (ESRF) (Beuvier et al., 2019). The equipment needed for the measurements of the thickness is an optical microscope, with a pair of polarizers, a condenser, a high resolution lens (X100 in our case) and a numerical camera. A precise calibration of the brightness of the microscope is required. The precision and stability of the microscope tuning constitute a limitation of the method : The light intensity, the diaphragm opening, the position of the condenser, and the exposure time of the camera, have to be strictly identical between the calibration and the analysis of the calcite crystal. Slight change on one of those parameters have important consequence on the results. Another limitation is that the measured light intensity is not linearly proportional to the thickness but follow a sigmoid (Beaufort et al., 2014;Bollmann, 2014) making difficult to estimate the thickness precisely at the two ends of the calibration. The use of standard polychromatic « white » light induce a small imprecision, because the temperature of light that depends on the microscope – some have a bluish light other have it more yellowish – will change slightly the result if not calibrated. There is a theoretical limit of the thickness estimation to about 1.56 μm when using a black and white camera. Some species have coccoliths thicker than this limit : in present ocean and Pleistocene sediments, rare examples are *Coccolithus pelagicus, Ceratolithus cristatus, Pontosphaera multipora* and coccoliths exceed this threshold only on limited surface of the thickest specimens. This threshold is achieved more commonly in the Paleogene for example for example with *Reticulofenstra bisecta,*or *Chiasmolithus grandis.* The estimation of calcite particles thicker than 1.56 μm needs to be done with a color camera with several calibration equations (Beaufort et al., 2014;González-Lemos et al., 2018). Here we propose a new method that solves those problems: the estimations are not the results of a calibration, they can be applied to crystals as thick as 1.7 μm, and are not dependent on the precise tuning of the light of the microscope.

## 3. Principles

The representation of the polarized light is based on Jones's calculus (Jones, 1941). The microscope is composed of two circular polarizers – one left oriented and the other right oriented – used alternatively and one circular analyzer.

*a.   Jones Matrices*

For an anisotropic material having its ordinary neutral axis horizontal, Jones matrix is given by


$$W_0 = T \begin{bmatrix} 1 & 0 \\ 0 & \eta e^{i(1-\phi)} \end{bmatrix}$$


where T is the (complex) transmission coefficient, $\eta$ is the diattenuation, and $\phi$ is the retardation,
with $\phi = \frac{2\pi}{\lambda} \Delta nd$ (where $\lambda$ is the wavelength, $\Delta n$ is the birefringence, $d$ is the thickness).

If the neutral axis is rotated by an angle $\theta$, the Jones matrix becomes
$$W_\theta = R(-\theta).W_0.R(\theta)$$

where $R(\theta)$ is the rotation matrix
$$R(\theta) = \begin{bmatrix} cos\theta & sin\theta \\ -sin\theta & cos\theta \end{bmatrix}$$


b.  *Proposed measurement scheme*

Assuming that $\eta = 0$ (no diattenuation), the input field is left-circularly polarized
$$P_L = \frac{1}{\sqrt{2}} \begin{bmatrix} 1 \\ i \end{bmatrix}$$

and the polarization analysis involved either a left circular polarizer made of a quarter-wave plate
at 45° followed by a horizontal polarizer
$$A_L = \begin{bmatrix} 1+i & 1-i \\ 0 & 0 \end{bmatrix}$$


or a right circular polarizer (made of a quarter-wave plate at -45° followed by a
horizontal polarizer)

$$A_R = \begin{bmatrix} 1+i & -1+i \\ 0 & 0 \end{bmatrix}$$


so that the measured intensities writes

$$I_{LL} = |A_L.W_\theta.p_L|^2 = |T|^2 sin^2 \left(\frac{\phi}{2}\right)$$

and
$$I_{LR} = |A_R.W_\theta.p_L|^2 = |T|^2 cos^2 \left(\frac{\phi}{2}\right)$$

c.  *Retrieving thickness*

One can see that $I_{LL}$ and $I_{LR}$ do not depend on the orientation $\theta$ of the neutral axes.
Moreover, the ratio
$$\frac{I_{LL}}{I_{LR}} = tan^2 \left(\frac{\phi}{2}\right) = tan^2 \left(\frac{\pi}{\lambda} \Delta nd\right), \tag{1}$$


does not depend on the transmission coefficient $T$.
In the case that we can assume that $\frac{\pi}{\lambda} \Delta nd < \frac{\pi}{2}$, implying that $d < \frac{\lambda}{2\Delta n}$,

then there is only one solution, $d$, to Eq (1) :
$$d = \frac{\lambda}{\pi \Delta n} arctan \left(\sqrt{\frac{I_{LR}}{I_{LL}}}\right) \tag{2}$$

Therefore the thickness can be estimated by grabbing two images of a thin calcite crystals, one
taken through a right circular polarizer ($I_{LR}$) and a second through a left circular polarizer ($I_{LL}$). $I_{LL}$
has a dark background and calcite crystals appear lighter. $I_{LR}$ has a light background and calcite
particles appear darker. They are negative images of each other (Fig. 1a). The ratio $\frac{I_{LR}}{I_{LL}}$ increases
with thickness (Fig. 1b). Applying Equation 2 to those two images gives the thickness and this
depends on the wavelength ($\lambda$) of the light used and the birefringence of calcite ($\Delta n = 0.172$).



**4. Material**
The methodology presented here was developed on a Leica DM6000 microscope, with a x100
objective having a numerical aperture of 1.47, and a condenser lens having a 1.2 numerical
aperture. Three circular polarizers made by Chroma Technology Corp. are integrated in the
microscope. (1) One right circular polarizer is positioned as analyzer. It consists of a linear
polarizer oriented at +90° placed below a quarter-wave plate oriented at +45° mounted in a Leica
cube and placed in the upper automatic turret of the microscope. This is a convenient place when
one wants to automatically remove this analyzer to use other filters. Alternatively, the analyzer can
be placed in its regular position.
Two polarizers are used alternatively when taking images of the same crystal : (2) a left circular
polarizer (LCP) consisting of a quarter-wave plate oriented at 45 followed by a linear polarizer
oriented at 0°, and (3) a right circular polarizer (RCP) made of a quarter-wave plate oriented at -
45° followed by a linear polarizer oriented at 0°.
If possible, the LCP and RCP are placed in the revolving filter chamber of the automated
condenser block. For a manual use, a quarter-wave plate could be placed under a linear polarizer,
and rotated manually from -45° (LCP) to 45° (RCP).
One of five monochromatic bandpass filters centered at 435, 460, 560, 655, and 700 nm
(AT435/20X, AT460/50M, ZET561/10X, AT655/30M and ET700/50M; all from Chroma Technology
Corp.) is positioned in the light trajectory after the light bulb. The 561 nm filter is used in routine
work because of its versatility (see below) and it is the one we recommend for a general use. The
other filters have been used in this study to test the method. In special occasions, we recommend
the use of a 700 nm filter to measure calcite particles with thickness ranging between 1.4-1.9 μm;
and 460 nm filter for detail measurements of thin particles in the range of 0.2-0.4 nm.
Two black and white numerical cameras are set up. A SpotFlex from Diagnostic Instrument, with
a CCD image sensor of 2048x2048 pixels that are 7.4 μm large. It is a 14-bit camera (16383 grey
levels in depth). And an Orca Flash 4.0 V2 from Hamamatsu, with a CMOS image sensor of
2048x2048 pixels that are 6.3 μm wide. It is a 16-bit camera (65548 grey levels in depth). The
tests of this method presented in results have been made with (i) surface sediment retrieved in the
Southern Pacific and spread onto a slide, and (ii) calcium carbonate crystals precipitated onto a
slide.
**5. Results**
To test the quality of the thickness estimations with the BCP method, the same field of view has
been studied in different light conditions (brightness, opening, and wavelength) and with different
cameras. In each condition, the two images $I_{LL}$ and $I_{LR}$ are captured and used to compute the
thickness $d$, with Equation 2. In some cases, in order to illustrate $d$, an image frame $d_i$ in 8-bits,
was computed using the following equation:
$$d_i = 256 \frac{d}{d_{max}} \tag{3}$$

where $d_{max}$ represents the maximum measurable thickness at a given wavelength. It is calculated
using the following equation:
$$d_{max} = \frac{\lambda}{\pi \Delta n} \cdot \frac{\pi}{2} \tag{4}$$

For calcite crystals, $d_{max}$ ranges between 1.17 μm at 405 nm and 2.03 μm at 700 nm.
*a.  Brightness*
The same field of view was captured at different exposure times with the SpotFlex camera.
Exposure time is the simplest way to change the brightness of an image. Figure 2 shows that the
fields of view captured at short exposure time (e.g., 5 ms) are extremely dark and conversely
those captured at long exposure time (e.g., 320 ms) are light with many saturated areas
(maximum Grey Level (GL) values). Except for those two extreme expositions (i.e., 5 ms and
320 ms) the GL values, in the resulting images in the bottom row of Fig. 2, are identical. In Fig. 3
the histograms of $I_{LL}$, $I_{LR}$ and $d$ are shown. At 320 ms the images are too light, and many areas
are saturated both in $I_{LL}$ and $I_{LR}$ and thus have the same GL values. Knowing that the solution of
Equation 2 is 0.81 µm when $I_{LL}=I_{LR}$ and $\lambda = 561$ nm, a spurious density peak appears in the
histograms at a thickness of 0.81 µm with an exposure time longer than 320 ms (Fig. 3). In areas
where $I_{LL}$ is saturated but not $I_{LR}$, the estimations are shifted toward thicker values, explaining
the thicker density pick found at 0.7 µm in the histogram of 320 ms (Fig. 3). The image
background, materialized in the histograms by the first peak, is around 0.1 µm for all exposures
but is shifted toward higher thickness up to 0.2 µm at 320 ms.
At 5 ms, the images are too dark to provide correct estimation of the background level (Fig. 3)
which, in turn, increases noise in the results. Therefore, in order to get correct thickness values, it
is important to avoid too low or too high brightness. Between those extremes light conditions, the
estimates of thicknesses are independent of brightness. To get the maximum depth details, it is
suggested to use the maximum light before saturation in $I_{LL}$, providing the largest range of grey
levels in both images and therefore a larger signal-to-noise ratio in the thickness estimates. In the
example given in Fig. 2, this maximum detail would be achieved between 80 ms and 160 ms.
The optical setting used in this experiment was not able to produce the darkest values (close to 1)
and lightest value (equivalent to 255 in 8-bit). The reason why those extreme values are not
reached is largely due to the imperfections of the circular polarizers that are composed of two
layers. Those imperfections are amplified at the extremes of the light ranges because of the
sigmoid shape of the thickness function (Fig. 1). In practice, the ratio $I_{LR} / I_{LL}$ is reached in the
flattest part of the sigmoids (Fig. 1b), for example between 0.10 µm and 1.41 µm with 561 nm light
wavelength. In consequence, the thickness measured in an empty part of the field of view was
0.10 µm at 561 nm when it should be 0. Also, the maximum measurable thickness is lower than
the maximum theoretical thickness: using a wavelength of 561 nm, we obtain a maximum of
1.45 µm of thickness instead 1.62 µm (Fig. 3).
*b.  Aperture*
The illumination tuning of the microscope is also important. The range of measurable thickness is
largest when the condenser is focused and centered following the Köhler illumination (Köhler,
1894). The more closed the field diaphragm is, the wider is the range of measurable thickness
(Fig. 4). Hence, both diaphragms (i.e., field and aperture) should be closed at their maximum in
order to maximize the range of measurable thickness.
*c.  Camera Type*
The two tested camera types (CMOS vs CCD; 14-bit vs 16-bit; different brand) produced the
same results. The same view field was captured with two different camera type without
measurable difference between the two resulting thickness images (Fig. 5).
The theoretical maximum measurable thickness ($d_{cmax}$) depends on the number of grey levels
($nGL$) achieved by the camera :

$$d_{cmax} = \frac{\lambda}{\pi \Delta n} arctan\left(\sqrt{\frac{nGL}{1}}\right)$$
(5)

At $\lambda = 561nm$, $d_{cmax}$ is 1.565 µm with an 8-bit camera, 1.622 µm with a 14-bit camera and
1.626 µm with a 16-bit camera. These $d_{cmax}$ are far above the maximum measurable thickness of
1.45 µm described in section 5.a. However, the low depth resolution of an 8-bit camera should
further limit the range of measurable thickness, although this was not tested here. Hence, both
14-bit and 16-bit can be used but we don't recommend to use 8-bit camera.

*d.* *Accuracy and Precision*
It is extremely difficult to estimate the measurement error in the present case because there is no
standard material for thickness comparison in the range of few nanometers. The thickness of the
wedge used to estimate the accuracy in González-Lemos et al. (2018) is measured at 250 nm
intervals which is not enough in our case. Also, its measurements are based on a birefringence
principle that is not strictly independent from our methodology. However, González-Lemos et al.
(2018) clearly validate the accuracy of birefringence method at 250 nm. The measurement of
coccoliths made by coherent X-ray diffraction (CXDI) at ESRF (Beuvier et al., 2019) requires the
use of silicon nitride ($Si_3N_4$) TEM windows influencing birefringence. Hence, those coccoliths
cannot be used later as standard. However, in this study, coccolith mass and size measurements
from the same culture using both birefringence and CXDI provide a comparison on statistically
similar results. The validity of the birefringence method is also demonstrated, although without
giving a value to the accuracy. The use of cylindric rods such as rhabdoliths (Beaufort et al.,
2014;Fuertes et al., 2014) is limited by the precision of the microscope used to produce the
measurement of their diameter, around 0.2 µm in our microscope, and likely due to issues with
natural variations in rhabdoliths (parts of which may be hollow). The BCP method does not use
any calibration, it is therefore theoretically absolute. It is accurate in the range given by the
inflection points in Fig. 1.
We determine the precision of the BCP method at the five different wavelengths by using the two
cameras on the same 7.74 µm transect of a *Pontosphaera japonica* (Fig. 6), producing 10 series of
measurements. At the difference with Fig. 5, and to produce feasible "user noise" we have slightly
shifted the focus and use different wavelengths. The root-mean-square error (RMSE) between two
series is used to determine the precision of the method. The RMSE ranges between 14 nm and
47 nm. The largest RMSE values result from largest focus differences and/or red colors (635 nm
and 700 nm). Best results were obtained at 561 nm and 435 nm with similar focus. When one
series of measurements was compared to the average of all the other series, the RMSE = 32 nm.
When it is limited to 435 nm to 561 nm, the RMSE = 12 nm. As we explain in detail in the next
section, longer wavelengths in red lower the precision. This is an order of magnitude smaller than
the spatial optical resolution which ranges between 150 nm and 240 nm in the present
microscopic setting at the 5 different wavelengths. The precision of the BCP method is expected
to be smaller in many cases. For example, the RMSE in the transect of Fig. 5 is 5 nm. The
difference of RMSE between Figs. 5 and 6 is related essentially to the focus that was well
reproduced in Fig. 5. The measurable masses of *P. japonica* in Fig. 6, is ranging from 65.3 pg to
69.9 pg with a standard deviation of 1.28 pg (N=10) and depends again, on the wavelength and
the focus.
*f.* *Wavelength and range of mesurable thickness*
The comparisons of the same transects captured at different wavelengths along an image frame
containing thick $CaCO_3$ particles emphasize the advantages and limits of each light wavelength.
The range of thickness measurable at a given wavelength is presented in Fig. 7. In the transects, a
plateau is reached at the maximum practical thickness (MPT) ; and when the particle thickness is
about 0.5 µm above the MPT, the thickness values decrease. It is not entirely clear why MPT is
about 84% lower than the maximum measurable thickness ($d_{max}$). This difference has been
described earlier (Bollmann, 2014). This discrepancy could be resulting from the quality of circular
polarizers used. The circular polarizers are made with polaroid filters that are not perfect and are
composed of two filters – a quarter-wave plate and a polarizer – creating some imperfections. As
an example, linear polarizers exhibit generally larger range of grey levels with darker background
than circular polarizers.
For the study coccoliths thicker than 1 µm like those of the Eocene, we recommend to use a light
with long wavelengths (e.g., red at 700 nm). On the contrary, for the study of thin coccoliths such
as most extant and Pleistocene species, we recommend to use shorter wavelengths (e.g., green
or blue). Short wavelengths reached a MPT at lower thickness but offer higher precision in the
measurement of the thickness and higher optical resolution permitting higher precision in the
measurement of the area. Plate 1a shows an *Emiliania huxleyi* coccolith, in which the slits, that are
present in the distal shield appears only in blue light. This illustrates an extreme cases, for which
the low wavelength has to be used to get a most precise thickness and mass measurements. The
distal shield of *E. huxleyi* is constructed with thin – ~100 nm – elements that do not touch each
other (Plate 1a). The detection of those elements above the background is extremely difficult
using wavelength at 700 nm but is possible using wavelength at 435 nm. In consequence, mass
measurements are underestimated at 700 nm because the distal shield is not completely
detected and producing a total area smaller than it is really (Table 1). Finally, this new method
cannot give accurate results for calcareous nannofossils) thickness above 1.7 µm like Cretaceous
*Nannoconus* species. For such material, we recommend to be critical with results close to MPT
and to use a color camera (Beaufort et al., 2014; González-Lemos et al., 2018) as in Fig. 7,
although less precise than the BCP method related to color calibration issues (González-Lemos et
al., 2018).

## 6. Protocol

1- Microscope setting : Köhler illumination done, diaphragms  as closed as possible, circular
polarizers (with a rotating quarter-wave plate or two circular polarizers : one left oriented and one
right oriented), circular analyzer, monochromatic filter,
2- Grab one image of a field of view with the circular polarizer oriented to the left (Image ILL)
3- Grab one image of the same field of view with the circular polarizer oriented to the right (Image
ILR)
3- Compute the image $d_i$ with equation 3 : $d_i = 256\frac{d}{d_{max}}$, with $d$ from equation 2 : $d =$
$\frac{\lambda}{\pi \Delta n} arctan\left(\sqrt{\frac{I_{LR}}{I_{LL}}}\right)$, and $d_{max}$ from equation 4 : $d_{max} = \frac{\lambda}{\pi \Delta n} \cdot \frac{\pi}{2}$
$d_i$ can be simplified in
$$d_i = 163 arctan\left(\sqrt{\frac{I_{LR}}{I_{LL}}}\right) \quad (6)$$


An example of a python routine that calculate the output image di is given here :
— — — — — — — — — — — — — — — — — — — — — — — — — — — — — — — — — —

```
# Import.Lib.
import sys
from PIL import Image
import math
from math import pi
# open Image file
img_ILL = Image.open(« /Path/image ILL.tif")
img_ILR = Image.open(« /Path/image ILR.tif")
# Create output image
img_d = Image.new(img_ILL.mode, img_ILL.size)
# Get image size
colomn,line = img_ILL.size
# Compute d for every pixel
for i in range(line):
for j in range(colomn):
ILL_val = img_ILL.getpixel((j,i)) + 1
ILR_val = img_ILR.getpixel((j,i))
# Compute thickness values
348                                 d = 163 * math.atan(math.sqrt(ILR_val / ILL_val))
```

```
# Ouptut image
img_d.putpixel((j,i), (int(d),))
# Show thickness image
img_d.show()
─────────────────────────────────────────────
```


4- Point measurement : di is an image that is scaled in grey levels and not in µm. In order to get
the thickness at one point (pixel) of an image, get at this position the grey level value, GL.

From equation (3) we obtain :

$$d = \frac{d_i . d_{max}}{256} \qquad (6)$$

$d_{max}$ is given in equation (4): For example for a calcite crystals ($\Delta n = 0.172$) and using a green
monochromatic light of $\lambda$ = 0.561 µm, $d_{max}$ is 1.63 µm. In that case GL must be divided by 160
in order to get the thickness at that point.


When one want to measure a particle (instead of a point) it may continue as follow :

5- Threshold : One must withdraw the background of the image without changing the GL values
of the particle. An easy way to do that is explained in the following ImageJ plugin. In this example
the maximum background GL value is 19  :

```
─────────────────────────────────────────────
run("Duplicate...", " ");
setThreshold(19, 255);
setOption("BlackBackground", false);
run("Convert to Mask");
run("Divide...", "value=255.000");
imageCalculator("Multiply create", "image.tif","image-copy.tif");
selectWindow("Result of image.tif");
─────────────────────────────────────────────
```


6- Average thickness ($\overline{d}$): To measure the lightness of the particle, select the region of interest
(ROI) containing an isolated particle. Measure the mean GL value of the ROI. Use equation (6) to
calculate the average thickness in µm of the particle.

7- Mass of the particle : $Mass = \overline{d}.a.\rho$ where a is the area in µm and $\rho$ is density of calcite in
pg/µm³ (=2.71).  the Mass is in picogram.

## 7.  Limits of Protocol

1-Thickness : As it was said earlier, this method is not applicable for particle thicker than the
practical dmax, that is 2µm using red light. This is not a strong limitation for coccoliths since most
of them are not thicker than 1.5 µm. In quaternary sediments, where the coccoliths are in majority
<1.2 µm thick, we prefer to use a blue color that gives the most precise results. When working
with Mio-Pliocene sediments, a green light is recommended because of large *Reticulofenestra*. In
Paleogene sediments it may be interesting to work with a red light.

2- v- units : BCP method is perfect for calcite crystals having their optical axis oriented
perpendicular to the light trajectory. During the crystallization of coccoliths, many crystals have
their optical axis radial oriented, the so-called r-units described by Young et al. (1992). Those
coccoliths (e.g Noelaerhabdaceae) are well measured by any polarization method including BCP.
In some species, the coccoliths have two types of crystals: those with optical axis oriented
radially ( r-units), and those with a vertical optical axis (v-units) (Young et al., 1992). The thickness
of crystals having a v-unit cannot be measured by birefringence methods. In some genus such as
*Pontosphaera* it does not impact significantly because the proportion of v-unit is limited. In some
genus such as *Coccolithus*, a larger proportion of the coccoliths is composed of v-units (the distal
shield), it is possible to use a correction factor as proposed by Cubillos et al. (2012). For
coccoliths composed exclusively of v-units such as the discoasters, BCP  and other birefringent
methods are not applicable.
3-Sample preparation : Most of the preparation method used in the study of fossil samples are
using glass as a support, whereas, some methods are using membrane with a small porosity (e.g.
0.45 µm) in order to retain the coccolith on it (see Giraudeau and Beaufort, 2007, for a review).
Such methods are classical used when studying living coccolithophore assemblages, the
collected sea water is filtered on a membrane that is subsequently mounted between slide and
coverslips with a mounting media that is sufficiently liquid to makes the membrane almost
transparent. Three types of membranes are used : Acetate cellulose, nitrate cellulose and
polycarbonate. The membranes are not completely transparent and this affect the measure of
thickness. To quantify this effect we mounted the same sample on glass only (GO), with
membrane on acetate cellulose (AC) and with polycarbonate membrane (PC).  The background
level measured in blue (560nm) was 14, 16, 19 GL with GO, AC and PC respectively : The
« opacity » of the membranes add 2 GL for AC and 5 GL for PC corresponding respectively to the
thickness of 11 nm and 26 nm or to mass/µm2 of 0.03 and 0.07 pg. These values are in the same
order of precision as expected with the BCP method. Because it is not possible to measure the
same object on the three types of support, we measure the average mass and thickness of
coccoliths from a large population belonging to the same species (E.huxleyi) in the same sample
replicates (MD97-2125; 5cm). We did not find any significant difference between the population
measured on the different supports (Table 3). There is no apparent limitation to measure calcite
thickness on membranes of that type. The small holes in the polycarbonate membranes are not
filled by the medium. They appear opaque observed in the microscope in both natural, and
circular polarized light (right and left). These holes can be seen by transparency through calcite
particles. In the BCP image projections, the holes do not appear prominently and they are half
darker and half lighter than background, inducing a small but significant noise in the resulting
thickness. Although this effect is not large, the use of this membrane is not recommended when it
is possible to use acetate cellulose membranes.

## 8. Conclusions

The alternative use of left and right circular polarization permits to measure the thickness of
calcite crystals in a universal manner without precise calibration of light. The BCP method has a
great advantage from previous methods for which it is difficult to maintain stable light (i) in time
(i.e., bulb aging, condenser vertical position,…) and (ii) in space since the field of view may not be
uniformly illuminated (i.e., low quality lens, uncentered condenser, ...). In all these situations, the
previously published linear or circular polarizer methods will provide different thicknesses
measurements whereas the BCP method described here will provide the same values. The choice
of the wavelength of the light used for the measurements is specific to a targeted thickness.
Thicker crystals will require longer wavelengths. Shorter wavelengths are recommended for
precise measurement of thin crystals. In practice, upper and lower limits of measurements
depend on the quality of polarizers and on the tuning of the microscope (Kohler illumination and
narrow diaphragms). With our microscope, the practical range of measurements is 84% of the
theoretical range. For example ,at 561 nm, the lower measurable thickness is 0. 10 µm and the
largest is 1.45 µm when theoretically the range should be 0 to 1.61 µm. It could be interesting to
test if other type of circular polarizers such as mineral ones could provide larger practical ranges.
The precision of the thickness measurements are an order of magnitude smaller – 0.012 µm to
0.030 µm – than that measurements of the length related to the resolution of an optical
microscope that is approximatively 0.20 µm using natural light.

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

**Acknowledgments**

The Fondation pour la Recherche sur la Biodiversité and the Ministère de la Transition Ecologique
et Solidaire are acknowledged for their support to the COCCACE project within the
program 'Ocean Acidification'. We thanks 2 anonymous reviewers for their constructive comments.
Captions :
**Figure 1**: **A.** Light intensity (arbitrary scale from min=0 to max=1000) going through a left circular
polarizer ($I_{LL}$) (top scale) or a right circular polarizer ($I_{LR}$) (bottom scale) associated with a left circular
analyzer in relation to the thickness of calcite crystals (birefringence Δn = 0.172), under
monochromatic light of wavelengths of 435 nm (indigo curve), 460 nm (blue curve), 561 nm (green
curve), 665 nm (red curve) and 700 nm (brown curve). **B.** Light intensities ratio ($I_{LR}/I_{LL}$) under
monochromatic light of the same wavelength as in A in relation to calcite crystals thickness.
**Figure 2**: Crops of images captured at different times exposure (in columns; 5 ms, 20 ms, 40 ms,
80 ms, 160 ms, 320 ms) in right circular polarization (first row; $I_{LR}$), left circular polarization (second
row; $I_{LL}$) and resulting thickness using Equations 2 and 3 with λ = 561 nm (third row; $d_i$). The resulting
thickness images are very similar in the range of time exposure.
**Figure 3**: Histograms (bins of 64 grey levels (top) and 6 nm (bottom)) of the same field of view as in
Fig. 2 , captured with green monochromatic light λ = 561 nm in right circular polarization (**top**), left
circular polarization (**middle**), and the resulting thickness using Equation 2 (**bottom**) at different
exposure times (black with plus signs: 5 ms, purple: 20 ms, light blue: 40 ms, blue: 80 ms, green:
160 ms and black with crosses: 320 ms).
**Figure 4**: Histograms (bins of 64 grey levels (top) and 6 nm (bottom)) of the same field of view as in
Fig. 2 , captured with green monochromatic light λ = 561 nm in right circular polarization (**top**), left
circular polarization (**middle**), and the resulting thickness using Equation 2 (**bottom**) at different
openings (Leica DM6000B scale ranging from 1 (closed) to 20 (open)) of the field diaphragm (black
with stars: 20, black with circles: 15, black with squares: 10, green: 8, blue: 5 and purple: 4).
**Figure 5**: A: Thickness along a transect (yellow line in the inset) measured with the Spotflex (red
line with crosses) and the Orca Flash cameras (blue line with plus signs). B: Relation between $I_{LL}$
(red), $I_{LR}$ (blue) and thickness (black) measurements made by the two cameras along the same
transect.
**Figure 6**: Precision of measurements made on the same 7.74 µm transect (yellow line in the inset)
across a *Pontosphaera japonica* (inset) with 2 cameras and at 5 or 3 wavelengths, producing
respectively 10 or 6 series of 129 points. Red: all wavelengths ($r^2$ = 0.996; RMSE = 0.032 µm); Blue:
435, 460 and 561 nm ($r^2$ = 0.994; RMSE = 0.012 µm). A. Relation between measure of a thickness
series compared with the average of all the others. the average thickness of 9 (or 5) series along a
transect and the thickness in the independent (not included in the average) series. The colored area
represents the 80% prediction bounds. B. Whisker plots of the residual, bars represent the
interquartile range, box represents the range between the 1st and 3rd quartiles. Standard deviation
= 0.032 (left in red) and = 0.019 (right in blue).
**Figure 7**: Thickness measurements made along two transects (T.1 in red and T.2 in white lines in
the left inset) of $CaCO_3$ crystals at 5 wavelengths (brown lines: 700 nm; red lines: 635 nm; green
lines: 561 nm; blue lines: 460 nm; indigo lines: 435 nm) and with polychromatic light grabbed by a
color camera (black lines; using the Hue values transfer function for thickness from Beaufort et al.,
2014 – this latter method allows measurement up to thickness of 4.5 µm after a complex calibration,
dotted black line is the thickness measured with the logit function in Beaufort et al., 2014, that
transfer GL in thickness values : note that for this image the white balance is not perfect). The 3
insets represent the images taken with a color camera (Spotflex) (left), a black and white camera
(Spotflex) at 700 nm (center) and the same camera at 435 nm (right). The maximum and minimum
measurements for each wavelength are indicated with an arrow.
**Plate 1**: Images of a coccolith of *Emiliania huxleyi* captured at wavelengths 435 nm (A) and 700 nm
(B). White bars are 1 µm long. Brightness has been adapted to enhance the contrast between
background and elements from the distal shield.
**Table 1** : Microscope parameters and inferred precision of the optics and measurements.
**Table 2** : Measurements at different wavelength of the coccoliths of *Emiliania huxleyi* presented in
Plate 1.
**Table 3** : Average morphology results of population of *Emiliania huxleyi* coccoliths measured on 3
different supports.


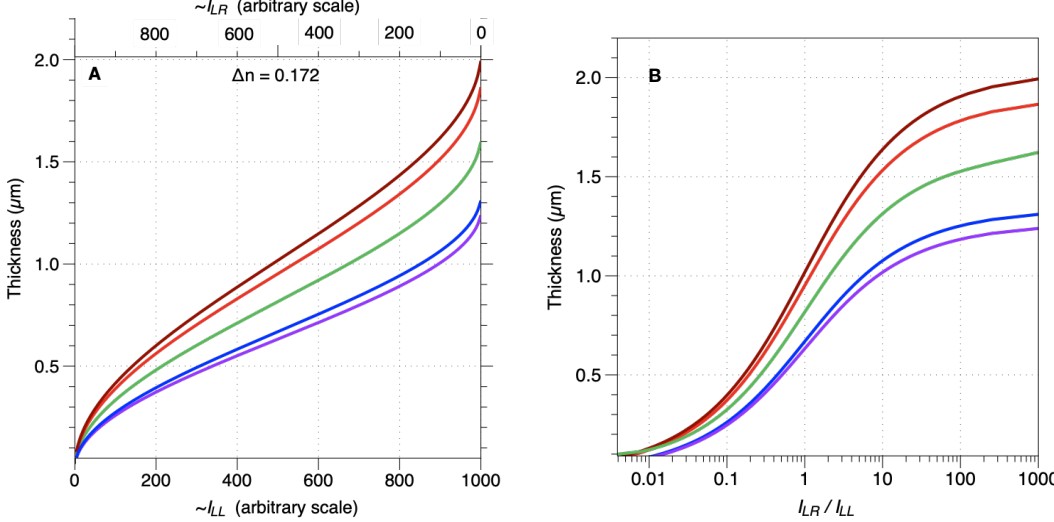

*Figure 1*



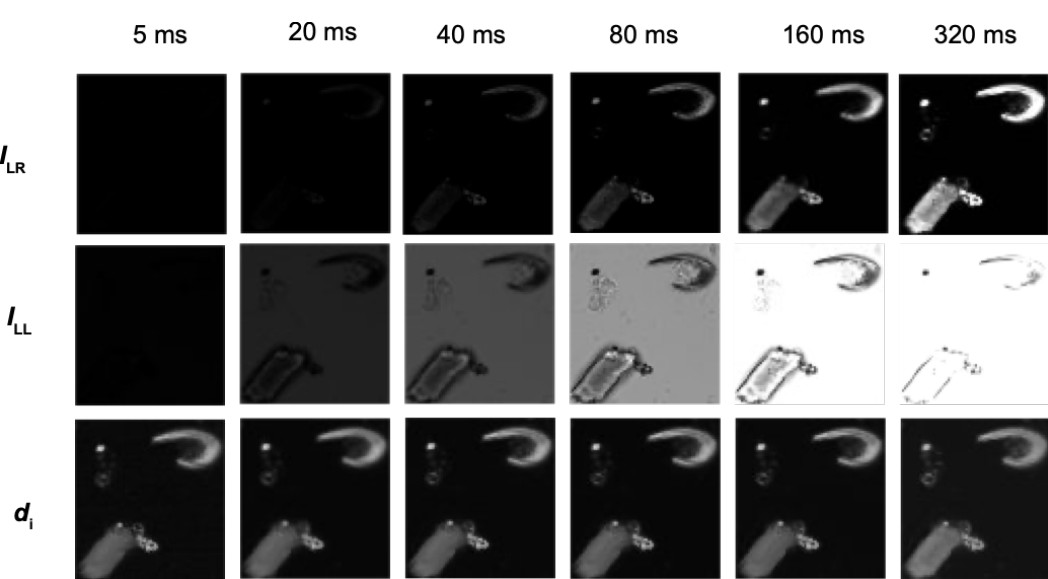

*Figure 2*


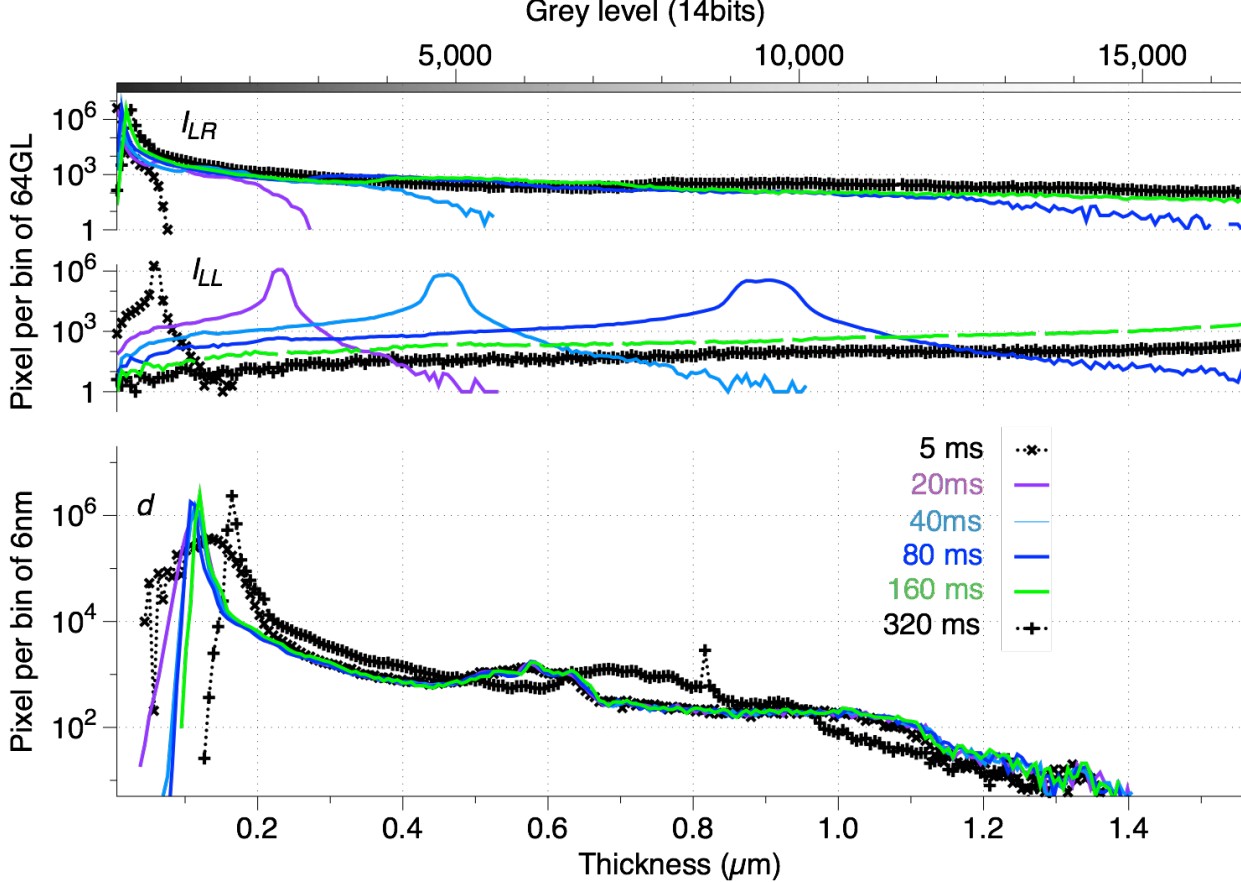

*Figure 3*

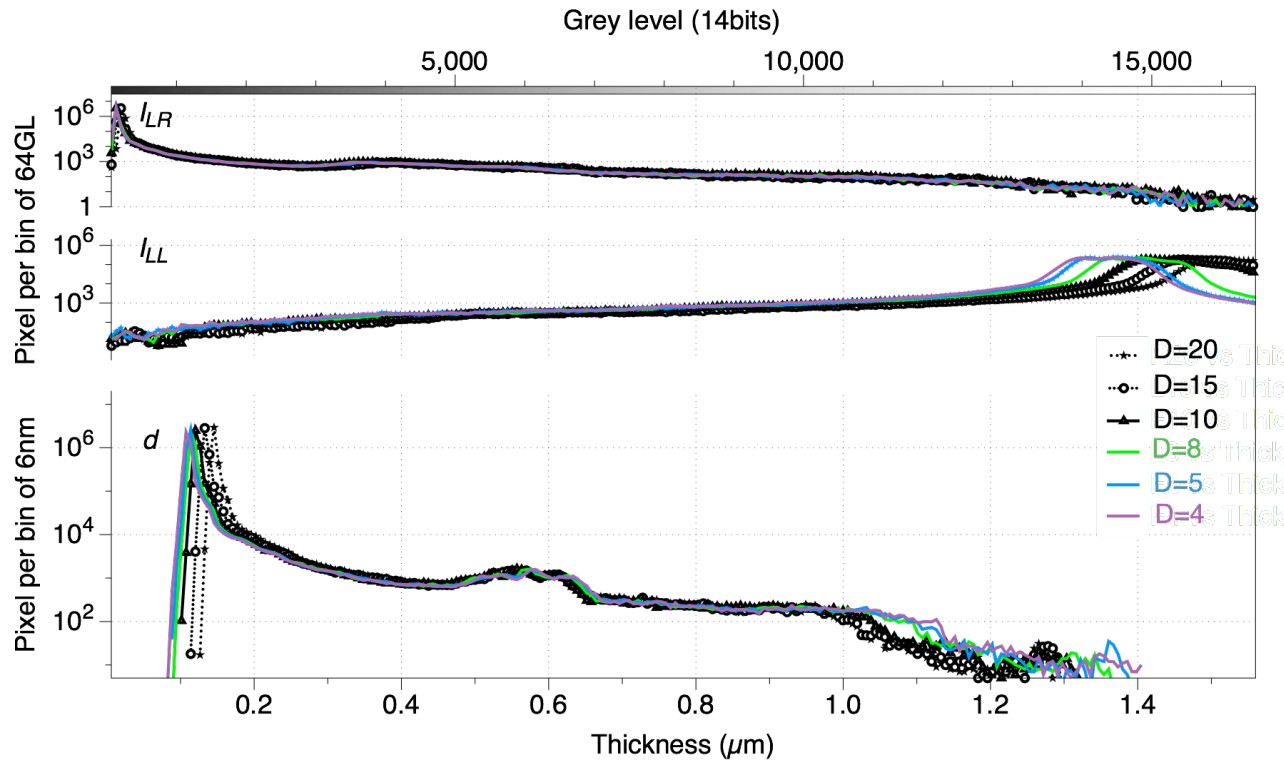

*Figure 4*


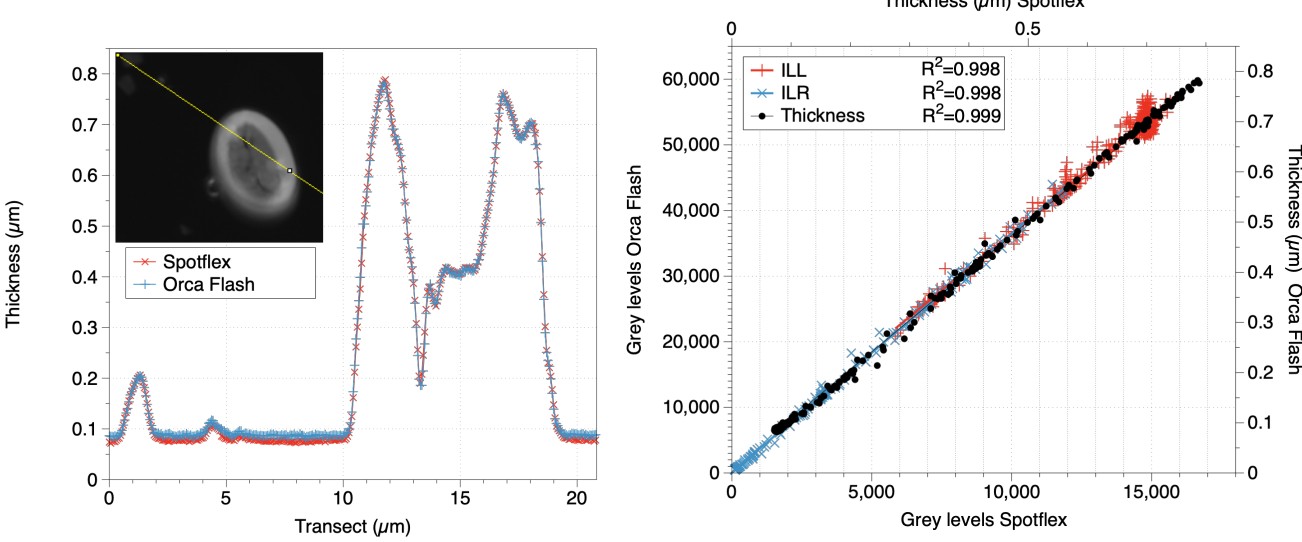

*Figure 5*


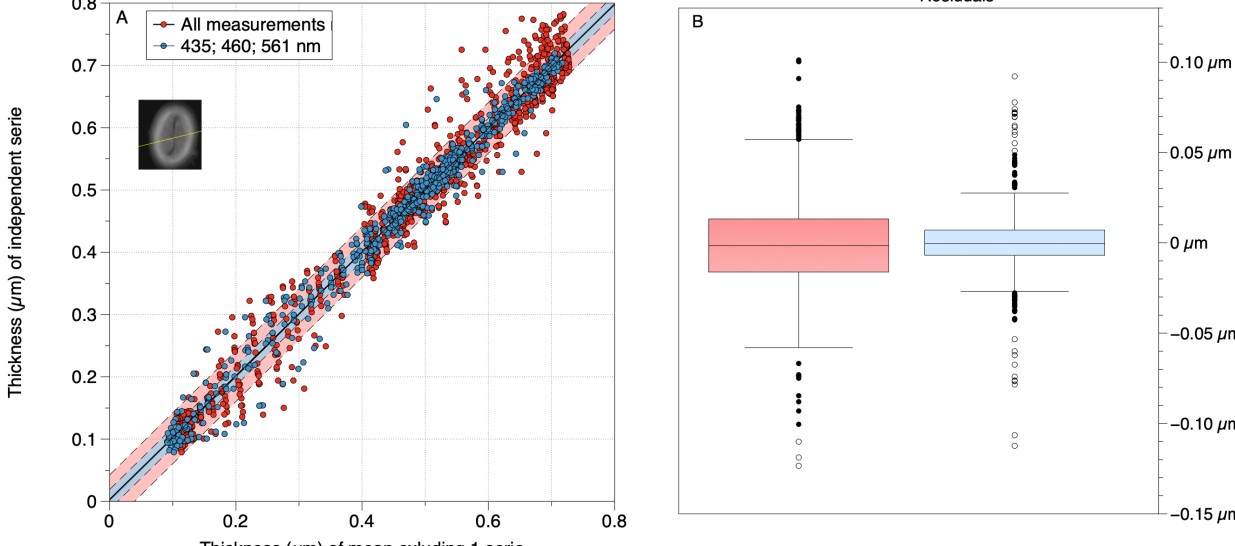

*Figure 6*


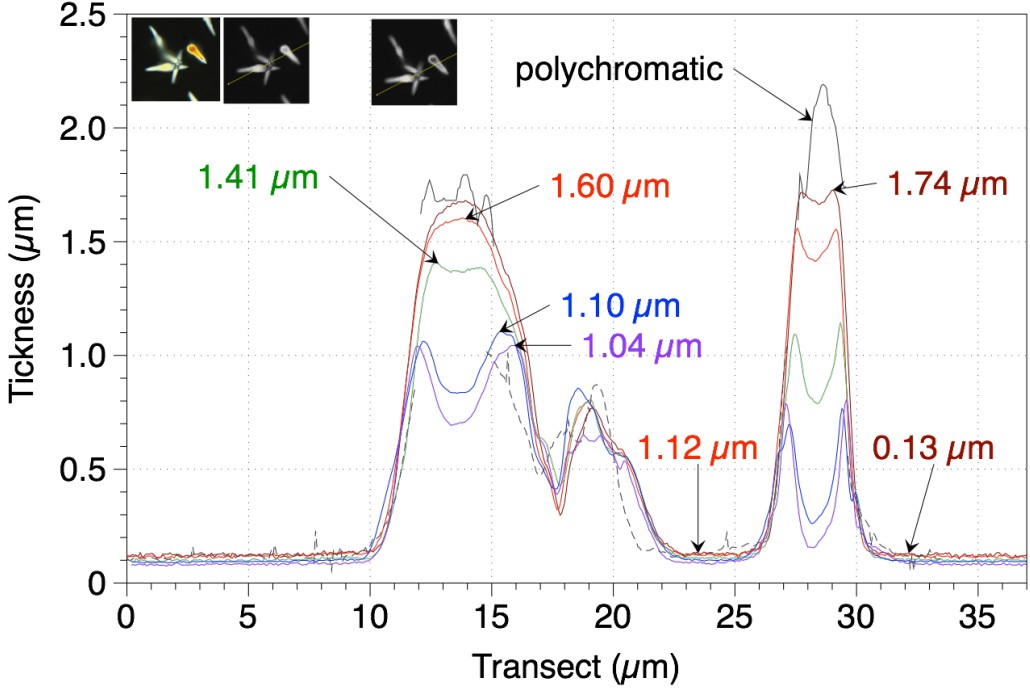

*Figure 7*


## Table 1

| Wavelength (λ) | Numerical apperture of lens | Numerical apperture condenser | Optical resolution | Maximum Measurable thickness | Theoretical thickness resolution (8bit) | Pratical thickness reproctily | Equivalent resolution |
|---|---|---|---|---|---|---|---|
| Equation / symbol | LNa | CNa | $λ/(2 * LNa)$ | $λ/(2 * 172)$ | $λ/(2*172*256)$ | RMCE | RMCE (µm)* |
| 435 nm (blue) | 1.46 | 1.2 | 0.148 µm | 1.26 µm | 4.9 nm | ~12 nm | 0.032 pg/µm |
| 460 nm (blue) | 1.46 | 1.2 | 0.156 µm | 1.34 µm | 5.2 nm | ~12 nm | 0.032 pg/µm |
| 561 nm (green) | 1.46 | 1.2 | 0.191 µm | 1.63 µm | 6.4 nm | ~12 nm | 0.032 pg/µm |
| 635 nm (red) | 1.46 | 1.2 | 0.223 µm | 1.85 µm | 7.2 nm | ~32 nm | 0.087 pg/µm |
| 700 nm (red) | 1.46 | 1.2 | 0.238 µm | 2.03 µm | 7.9 nm | ~32 nm | 0.087 pg/µm |

## Table 2

| Lambda nm | Mass (pg) | Area (µm2) |
|---|---|---|
| 435 | 4.43 | 7.97 |
| 460 | 4.23 | 7.94 |
| 561 | 4.30 | 7.94 |
| 635 | 3.97 | 7.12 |
| 700 | 3.96 | 6.53 |

## Table 3

| MD97-2125 (5cm) | Nucleopore | Acetate Celluleose | Glass |
|---|---|---|---|
| Mass (pg) | 1.66 pg (0.94 std)) | 1.78 pg (0.93 std) | 1.79 pg (0.70 std) |
| Thickness (µm) | 0.24 µm (0.05 std) | 0.25 µm (0.09 std) | 0.23 µm (0.04 std) |
| Number | 90 *E.huxleyi* | 168 *E.huxleyi* | 1285 *E.huxleyi* |


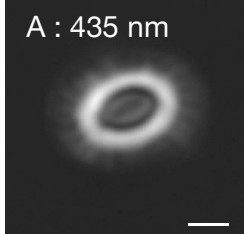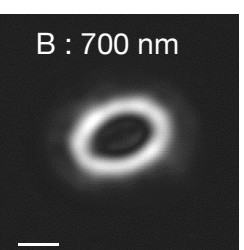

*Plate 1*