# Peer review of "Technical Note : A universal method for measuring the thickness of microscopic calcite crystals, based on Bidirectional Circular Polarization"

_Biogeosciences, 2020_

## Referee Comment (RC1) · Anonymous Referee #1 · 26 May 2020

The authors present a new method that tackles a key limitation of current methods that use light microscopy to determine the thickness of coccoliths – that thickness measurements based on grayscale (or colour) images are very specific to the exact set up of the microscope and are therefore very difficult to replicate between labs. The new method removes the microscope and camera settings from 'the equation' as it were, and instead leverage optical physics to base their calculation of thickness on the difference in (grayscale) intensity between a left circular polarizer and a right circular polarizer, a novel solution to this challenge. I have minor comments and would

recommend acceptance after minor revisions.

Overall, the manuscript is succinctly written and, whilst this is a technical paper, I think that the reach of this method to the broader coccolithophore and mineralogical community would be greater if the introduction included slightly more context on the crucial importance on coccolith carbonate production in the Earth system and particularly the role that estimating coccolith calcite has in those calculations (for example, even the statement that thickness is required to accurately calculate mass). The introduction should also highlight briefly that black and white birefringence colours to the thickness limit of ∼1.55 um is quite widely applicable to the most dominant extant and Pleistocene species of coccolithophores but is not widely applicable for species with larger and more heavier coccoliths earlier in the Neogene and Paleogene (which may not be obvious to someone less familiar with previous technical papers on these thickness methods – line 200 for instance is perhaps slightly misleading when larger Coccolithus coccoliths, for example, could exceed this thickness limit even in more recent sediments).

Whilst the method well describes the microscope and camera set up, and the optical physics underlying the thickness calculation in this new methods, I felt it is missing a description of the step between taking the images and calculating the thickness, for instance which image software has been used and how the software is set to determine the ratio ILR/ILL for each pixel. Without this additional description it might be challenging for the reader to set up this method using their own equipment. For completeness, a statement about any limitations of this method for species that are not fully birefringent under circular polarised light (e.g., Discoasters or even Pontosphaera japonica, which is used as the illustrative coccolith here) and any additional steps for applying this method when using culture samples that are therefore on filters (i.e., a very brief comment on whether the background will be normalised automatically when determining the ratio ILR/ILL and how this might affect the wavelength of light used to achieve optimal brightness – this might fit better in the discussion section on this

part). A couple of other details could also be included in the Material section – what is the resolution of the microscope at each wavelength, the numerical aperture of the objective and condenser, and the camera resolution (can be calculated with some of the details you provide but it would be easier just to state it).

Typically, similar thickness methods have been tested with combinations of quartz wedges or increasing thicknesses of retardation materials. Of course, this has been necessary for calibration, which is not required in this method. However, I wonder if there was a reason why there is no direct comparison presented between the new method and previous methods and the thickness of the samples. Although these calibration materials often quickly exceed a thickness of 1.55 um over a small spatial area, if those are the pre-existing calibration approaches then a direction comparison between the results would have been useful. This particularly would be necessary if your sample includes a mix of coccoliths where the 1.55 um thickness limit is likely to be exceeded and you would need to use a colour birefringent approach for some specimens and therefore have to calibrate your microscope to those settings too (as in Figure 7).

Both of the cameras you have used are black and white cameras – are the results different if you use a colour camera set up throughout? What other steps would you need to be aware of in this case? This would be important assuming that most groups would need to have the capacity for measuring also at thicknesses greater than 1.55 um and might need to be quite flexible to measure a range of thickness. In Figure 7, you do contrast the colour camera with different wavelengths imaged using the black and white camera. However, there is no comparison of the colour camera used at thickness below ∼1.5 um.

In mixed assemblages, there are obviously many species of many different coccolith sizes and thickness. Here, you suggest that using longer wavelengths for thicker coccoliths and shorter for thinner coccoliths to optimise optical resolution and contrast. In practice, do you therefore need to image every field of view at a range of wavelengths to ensure that you can get the most precise thickness measurements for all

the species in your samples, unless it's a really monospecific assemblage or a culture sample? Would you recommend that groups default to a (more complex calibration) for a colour set-up unless they will only be looking a more recent sediments or small, extant cultures?

In general, the succinct nature of the manuscript does assume that the reader is already very familiar with previous developments in thickness measurement methods. Whilst of course, this doesn't need to be described in huge detail again, I think at the moment that most readers would need to read several previous papers to get a good understanding on this new method within the context of what has gone before. Therefore, I would recommend that the authors further develop some of the statements in their introduction, method and discussion sections to provide slightly more context about the development of ideas and approaches in this technical area.

General comments on manuscript presentation: Some careful proofreading of the text will be necessary before the final manuscript is submitted, as there are minor typos and grammatical errors throughout that in some instance obscure the easy reading of the text. Also, there are minor inconsistencies in figure presentation (for examples A and B written on Figure 1 and Figure 6 but not on Figure 5) and minor layout adjustments that would improve presentation (for example, inconsistent size and alignment of the three microscope images inset in Figure 7, poor alignment of part A and B in Figure 6, a redundant Residuals title, and an unnecessarily small inset coccolith image in the same figure, poor alignment of parts A and B in Figure 1).

---

## Referee Comment (RC2) · Anonymous Referee #2 · 16 Jul 2020

The presented method by Luc Beaufort and colleagues is a very innovative and welcome "upgrade" to the existing approaches to "weighing coccoliths". The method and its results are well illustrated and theoretically correctly described – and I am convinced that this is a major step forward, as it will eliminate some of the "calibration issues" as encountered and discussed in previous literature. This technical note should be ready for final publication after minor revisions that focus on improving the introduction, adding an introductory "visual" of the general principles of the method (and "shopping list" Table, possibly) and fixing grammatical issues.

A technical note it is. The language is indeed rather technical. I believe the authors could help the non-expert reader – but potential future user? – by revising the Introduction and providing easier access to the background of this methodology and its development – and most importantly, explain why we need such a method – what research questions will be better addressed because of it?

The other reviewer mentions a similar need for such revisions, and commented that the English language needs edits. I apologize for the tardiness of this review (a myriad of excuses are applicable), but hope with the detailed comments below, that part of these language issues can be quickly remedied.

I wrote these comments as I went through the text, so please forgive the "less ordered" structure thereof.

Abstract First sentence – do you mean to say "Coccoliths" (the products) or "The coccolithophores" (the organisms) – I'd remove "The" before "coccoliths" if the former. Also in the next sentence. They weigh "a few picograms".

"However, the current method"

Apparatus = equipment (change also elsewhere)

"More precisely, . . . —- the calcite crystal."

Line 15: brands (plural)

Line 21: not sure if "rotative" is the right term. Rotational?

Line 27: "Alternatively, the birefringence characteristic of coccolith in polarized optical microscopy has been used to estimate their mass" → . . . characteristics of coccoliths . . . have been used . . .

line 27-28: expand on this background, explain for the reader who is not familiar with this literature. What have been the major steps in the development of this method/approach so far? That will make the bridge to why this approach is so much
better (easier and more accurate, avoid calibration issues and equipment differences).

"This method is rapid and precise. The camera sensor produces excellent measurement of light. The camera sensor measures the light that had come through the polarizers and a calcite crystals to convert into a thickness value."

– These are a set of rather short statements – which are not wrong, but they read like a bulleted list for a talk with main points, rather than a ease-the-reader into the topic introduction. – Grammar issues: "The camera sensor measures the light that travels through the polarizers and calcite crystals, which is converted into a thickness value"

Line 31: "The estimation made" – you mean, the thickness estimation? Line 32: "One of its limitations" – may need to start this sentence with a "However," as it is nuancing the statement before.

A lot of technical microscopy and camera terms are thrown in at the onset. Would be better with some general statement of the "equipment needed" in more general terms first? I'd keep the technical parts (condenser, exposure time etc.) for later in the method description – rather than going there immediately in the current introduction. Suggestion: create a section "Previous applications and limitations" or some header like that, before your current section 2. Principles (not to confuse it with your "3 Material" section – this should cover the range of settings used so far in the different labs)

Focus the Introduction on the reasons why one would need accurate measurements of coccoliths. Why do we bother? How "inaccurate" or "accurate" have previous investigations / measurements been, or how comparable between labs, and what type of answers have we gained from weighing coccoliths thus far? Why do we need a good calibration, or find ways to avoid such calibrations (where your new method comes in as the best alternative so far)?

Line 35: "Another limitation is that the measured light intensity is not linearly proportional to the thickness but follow a sigmoid (Beaufort et al., 2014; Bollmann, 2014)

making difficult to estimate the thickness precisely at the two ends of the calibration".

This statement could be illustrated with a Figure – refer to Figure 1 for example, or create an "easy entry" graphic before current Figure 1 – so that the more inexperienced reader /user can follow the rationale of the "two ends of the calibration" (including what values/units are at the "ends of the calibration"?)

"Here we propose a new method that solve those problems" – a new method that solves (singular verb conjugation) – please check throughout for such verb conjugation (plural vs. singular).

Line 45: "The representation of the polarized light is based on Jones's calculus" I don't actually know if I understand this sentence. Well, I don't ;-)

Line 81: "two images of a thin calcite crystals, taken one through a right circular polarizer and a second through a left circular polarizer:"

change to "one taken through . . . and another through" or (active) "taking one and . . . a second / another . . ."

line 92: "If this not necessarily the analyzer can be placed in its regular position" fix this sentence structure ("is not necessary"). Not sure what you mean to say. Is "this" referring to "use of other filters"?

Section 3 Material reads as a listing of microscopy parts – maybe consider a "shopping list" Table instead? And in the text comment on where the parts are placed relatively to each other, or explain more the rationale what the combination of parts brings.

For example line 93 "One (and one) left (right) circular polarizer, LCP (RCP), made of a quarter-wave plate oriented at 45° (-45°) followed by a linear polarizer oriented at 0°." Is not really a stand alone sentence –

Instead, consider [RATIONALE OF STORY FLOW IN BRACKETS] : "Two polarizers are used alternatively when taking images of the same crystal [WHAT IS THE PUR-

POSE]. One left circular polarizer (LCP), made of a quarter-wave plate oriented at 45°, and one right left circular polarizer (RCP), oriented in the opposite direction (-45°) [WHAT SHOULD WE BUY]. In both settings, a linear polarizer oriented at 0° is applied [PLEASE EXPLAIN ITS POSITION RELATIVE TO THE CIRCULAR POL – maybe an overview diagram would be helpful]."

Line 99: "The other filters are used to test the method [YOU MEAN EVERY USER SHOULD DO SUCH TESTS; OR ONLY IN CASE OF THIS PARTICULAR METHOD DEVELOPMENT YOU REPORT ON HEREIN?] and in special occasions when study of relatively thick calcite particles in the range of 1.4-1.9 $\mu$m in the case of 700 nm" the latter part of the sentence should read (as a separate sentence?): "In special occasions, a 700 nm filter could be used to measure relatively thick calcite particles (1.4-1.9 $\mu$m range)."

"Diagnostic Instruments" is the company's name.

4. Results

line 113: "where ðÍŚŚmax represent the maximum measurable thickness" → "represents" (singular verb conjugation)

section 4.1 Lightness — I think the more appropriate technical term is "Brightness" (change throughout; also "light" → "bright")

line 117: "at different time exposures" → "at different exposure times"

line 120: "Except for those two extreme expositions (i.e., 5 ms and 320 120 ms) the GL values are identical" – you mean the GL levels in the composite ("combined LCP and RCP", di, bottom row) images are identical – because they clearly are not identical for the series of exposure times for the individual CP images. You need to explicitly state that here.

Line 129: "the estimates of thicknesses are independent from the lightness" → "the thickness estimates are independent of brightness"

Line 130: "providing the highest range of grey levels in both images". Could you please explain why that is so (in short, easy terms the non expert can understand).

Line 136: "In consequence, the thickness measured in absence of ANY particle was 0.10 $\mu$m at 561 nm when it should be 0" - or say "empty part of the slide" for clarity.

Line 142: "More the field diaphragm is close, wider is the range of measurable thickness (Fig. 4)." Fix grammar – "The more closed the field diaphragm is, the wider is the range of measurable thickness".

Line 146: "produced the same measure" – produced the same results.

Line 153: "an 8-bit camera should further limits the range of measurable thickness" – "should further limit"

Line 161: "requires the use" Line 162: "coccoliths mass and size measurements" – "coccolith mass and size . . ."

Line 165: "The use of cylindric rods such as rhabdoliths (Beaufort et al., 2014;Fuertes et al., 2014) is limited by the precision of the microscope used to produce the measurement of their diameter, around 0.2 $\mu$m in our microscope" –

and likely due to issues with natural variations in rhabdoliths (parts of which may be hollow).

Line 170: "we voluntarily did not produce identical focus and use different wavelengths in order to produce generalized values" – you mean "on purpose", "purposefully" – to produce feasible "user noise"?

Line 171: "When the wavelength are separated, the 5 RMSE range between 14 nm and 47 nm" – wavelengths (plural); but more importantly I don't understand what you try to convey in this statement.

Line 181: change to "measurable"

[Figure]

Line 194: "The distal shield of Emiliania huxleyi coccoliths illustrate well an extreme measurement cases where the lower wavelength has to be used to get a precise thickness and mass measurements" what do you mean, "an extreme measurement cases" – suggest to rephrase to "For example, the lower wavelength has to be used to get precise thickness and mass measurements of the very thin distal shields of Ehux"

Emiliania huxleyi should be in italics.

Line 200/201: "close to the MPT", "although this is less precise . . . due to some calibration issues [maybe shortly comment on what calibration issue in this case?]."

Line 207: "In all situations mentioned above" – "In all these situations" (you mention them in the previous sentence, not "above")

Line 210: Thicker crystals (plural) Line 212: closed or "narrow" diaphragms?

I think all figures are very well designed, and informative - they are needed to illustrate the rather technical written descriptions.

Figure 5: I think the rendition of these profiles across the imaged coccolith is a very effective way for the non-expert to "get" the method. Maybe integrate such "profile" also in a first (new/additional) Figure that you may add to explain the basic principles in a diagram (microscope equipment, imaging and brightness-thickness conversion principle) in support of your Introduction.

Check to replace "lightness" »> "brightness" also in figure captions.

---

## Author Comment (AC1) · 10 Sep 2020

We are thankful for the referee's comments that are generally positive and constructive. We answer his/her different comments in blue inside the text.

The authors present a new method that tackles a key limitation of current methods that use light microscopy to determine the thickness of coccoliths – that thickness measurements based on grayscale (or colour) images are very specific to the exact set up of the microscope and are therefore very difficult to replicate between labs. The new method removes the microscope and camera settings from 'the equation' as it were, and instead leverage optical physics to base their calculation of thickness on the difference in (grayscale) intensity between a left circular polarizer and a right circular polarizer, a novel solution to this challenge. I have minor comments and would recommend acceptance after minor revisions.

Overall, the manuscript is succinctly written and, whilst this is a technical paper, I think that the reach of this method to the broader coccolithophore and mineralogical community would be greater if the introduction included slightly more context on the crucial importance on coccolith carbonate production in the Earth system and particularly the role that estimating coccolith calcite has in those calculations (for example, even the statement that thickness is required to accurately calculate mass).

Yes, we agree and we will write a longer and more practical introduction.

The introduction should also highlight briefly that black and white birefringence colours to the thickness limit of ~1.55 um is quite widely applicable to the most dominant extant and Pleistocene species of coccolithophores but is not widely applicable for species with larger and more heavier coccoliths earlier in the Neogene and Paleogene (which may not be obvious to someone less familiar with previous technical papers on these thickness methods – line 200 for instance is perhaps slightly misleading when larger Coccolithus coccoliths, for example, could exceed this thickness limit even in more recent sediments).

Yes, we agree. Some *C.pelagicus* or *S. apstenii* may be thicker than 1.55µm. We will discuss that in the introduction and also in the discussion because this problem can be partly solved by using red color.

Whilst the method well describes the microscope and camera set up, and the optical physics underlying the thickness calculation in this new methods, I felt it is missing a description of the step between taking the images and calculating the thickness, for instance which image software has been used and how the software is set to determine the ratio ILR/ILL for each pixel.

OK. We will explain in the new version how to use ImageJ to do that with some code lines of a plugin.

Without this additional description it might be challenging for the reader to set up this method using their own equipment. For completeness, a statement about any limitations of this method for species that are not fully birefringent under circular polarised light (e.g., Discoasters or even Pontosphaera japonica, which is used as the illustrative coccolith here) and any additional steps for applying this method when using culture samples that are therefore on filters (i.e., a very brief comment on whether the background will be normalised automatically when determining the ratio ILR/ILL and how this might affect the wavelength of light used to achieve optimal brightness – this might fit better in the discussion section on this part).

We will speak of the problem of no birefringent form such as discoasters. This is an important point. The birefringence of the filters used in culture or seawater samples can be limited and in our experience, the brightness of the membrane is not 'additive' to the coccosphere certainly because of the focus point of the coccosphere is few micron above the membrane. We will add a section on this problem using the same sample prepared on membrane and on glass to show the effect of the membrane.

A couple of other details could also be included in the Material section – what is the resolution of the microscope at each wavelength, the numerical aperture of the objective and condenser, and the camera resolution (can be calculated with some of the details you provide but it would be easier just to state it).

We have given the resolution of the microscope at each wavelength in Table1.  We agree it comes late in the text. The table is also not come clear enough. So we will place a table with those parameters in the methodological part.

Typically, similar thickness methods have been tested with combinations of quartz wedges or increasing thicknesses of retardation materials. Of course, this has been necessary for calibration, which is not required in this method. However, I wonder if there was a reason why there is no direct comparison presented between the new method and previous methods and the thickness of the samples.

In the introduction we discussed shortly the reason why we did not use a wedge to calibrate the BCP method : The main reason is that the wedges are not precise enough. The resolution of the BCP method is about ten times more precise than the wedge calibration, and therefore we cannot make a comparison.

Although these calibration materials often quickly exceed a thickness of 1.55 um over a small spatial area, if those are the pre-existing calibration approaches then a direction comparison between the results would have been useful. This particularly would be necessary if your sample includes a mix of coccoliths where the 1.55 um thickness limit is likely to be exceeded and you would need to use a colour birefringent approach for some specimens and therefore have to calibrate your microscope to those settings too (as in Figure 7).

We can use a red colored filter to increase the thickness for this. This is discussed in the manuscript. Again the wedge is not precise enough in this matter and the retardation method would not be really independent. We will discuss in depth this problem in the discussion section.

Both of the cameras you have used are black and white cameras – are the results different if you use a colour camera set up throughout? What other steps would you need to be aware of in this case? This would be important assuming that most groups would need to have the capacity for measuring also at thicknesses greater than 1.55 um and might need to be quite flexible to measure a range of thickness.

This is an important point that we will explain a bit more in the text. We have also used a color camera. The results are exactly similar, because the method is based on monochromatic light for which a color camera is useless. Monochromatic light is an important requirement. Therefore the color issued from birefringence is not expressed in terms of color, but in black and white. Our eyes and a color camera will see a blue image using a blue filter, a red image with a red filter… It is possible to show this with a figure. This is partly shown in Figure 7 that shows images of thick calcite crystals with a color camera and the black and white correspondence with two types of monochromatic light.

In Figure 7, you do contrast the colour camera with different wavelengths imaged using the black and white camera. However, there is no comparison of the colour camera used at thickness below ~1.5 um.

We did not want to confuse the reader: the thickness of crystals smaller than 1.55 μm cannot be estimated with the color method because those crystals are gray. We will provide the thickness estimates less than 1.55 μm with the color camera with a dashed line, and given the explanation in the caption.

In mixed assemblages, there are obviously many species of many different coccolith sizes and thickness. Here, you suggest that using longer wavelengths for thicker coccoliths and shorter for thinner coccoliths to optimise optical resolution and contrast. In practice, do you therefore need to image every field of view at a range of wave- lengths to ensure that you can get the most

precise thickness measurements for all the species in your samples, unless it's a really monospecific assemblage or a culture sample? Would you recommend that groups default to a (more complex calibration) for a colour set-up unless they will only be looking a more recent sediments or small, extant cultures?

We agree that in the case of huge difference in thickness of species, one could be tempted to do two scanning at two different wave lengths. In our experience, this is not required. We regularly work with sediment covering the last 25 Ma without problems.

In general, the succinct nature of the manuscript does assume that the reader is al-ready very familiar with previous developments in thickness measurement methods. Whilst of course, this doesn't need to be described in huge detail again, I think at the moment that most readers would need to read several previous papers to get a good understanding on this new method within the context of what has gone before. Therefore, I would recommend that the authors further develop some of the statements in their introduction, method and discussion sections to provide slightly more context about the development of ideas and approaches in this technical area.
General comments on manuscript presentation: Some careful proofreading of the text will be necessary before the final manuscript is submitted, as there are minor typos and grammatical errors throughout that in some instance obscure the easy reading of the text. Also, there are minor inconsistencies in figure presentation (for examples A and B written on Figure 1 and Figure 6 but not on Figure 5) and minor layout adjustments that would improve presentation (for example, inconsistent size and alignment of the three microscope images inset in Figure 7, poor alignment of part A and B in Figure 6, a redundant Residuals title, and an unnecessarily small inset coccolith image in the same figure, poor alignment of parts A and B in Figure 1).

We will do our best to extend introduction, method and discussion. And process a careful proofreading of our manuscript.

---

## Author Comment (AC2) · 10 Sep 2020

We thank the reviewer for these comments and for the corrections made to the text. We will follow up on all the points he or she has made.

---

## Author Response (AR1)

AIX-MARSEILLE-UNIVERSITÉ UM34 – CNRS UMR 7330
BP 80
AVENUE LOUIS PHILIBERT
13545 AIX-EN-PROVENCE CEDEX 4
TÉL : 04 42 97 15 00 – FAX : 04 42 97 15 05

Luc BEAUFORT
Directeur de Recherche CNRS
beaufort@cerege.fr
33 4 42 97 15 71

Aix en Provence, le 6 novembre 2020

Dear Lennart de Nooijer,

Here we submit the revisions of our manuscript. We corrected the manuscript closely following the comment and recommendation of the two reviewers. We exactly did what is written in our 2 rebuttal letters. In the manuscript file the changes are highlighted in blue.
We are sorry to have been so slow to produce those corrections. Hope it is not a problem.

Thanks for your editing,

Best regards,

On behalf of the coauthors,

Luc